Cathepsin B plays a role in spermatogenesis and sperm maturation through regulating autophagy and apoptosis in mice

Wen Zongzhuang 1
Zhu Haixia 2
Wu Bin 3
Zhang Aizhen 3
Wang Hongxiang 2
Cheng Yin 2
Zhao Hui 1
Li Jianyuan 4
Liu Min lm@sdfmu.edu.cn 1
Gao Jiangang jggao@sdfmu.edu.cn 1 2
1 Medical Science and Technology Innovation Center, Shandong First Medical University , Jinan , China
2 School of Life Science and Key Laboratory of the Ministry of Education for Experimental Teratology, Shandong University , Jinan , China
3 Department of Reproductive Medicine, Jinan Central Hospital, Cheeloo College of Medicine, Shandong University , Jinan , China
4 Key Laboratory of Male Reproductive Health, Institute of Science and Technology, National Health Commission , Beijing , China
Svingen Terje
Electronic publication date: 2022 Dec 2
Publication date: 2022
Volume: 10
Electronic Location ID: e14472
Received 2022 Aug 9; Accepted 2022 Nov 6
Copyright: ©2022 Wen et al.
Copyright year: 2022
Copyright holder: Wen et al.
License: This is an open access article distributed under the terms of the Creative Commons Attribution License, which permits unrestricted use, distribution, reproduction and adaptation in any medium and for any purpose provided that it is properly attributed. For attribution, the original author(s), title, publication source (PeerJ) and either DOI or URL of the article must be cited.
License URL: https://creativecommons.org/licenses/by/4.0/

Keywords: CTSB, Spermatogenesis, Sperm maturation, CRISPR/Cas9

Funding: National Key Research and Developmental Program of China 2018YFC1003602 This work was supported by the National Key Research and Developmental Program of China (Grant No. 2018YFC1003602). The funders had no role in study design, data collection and analysis, decision to publish, or preparation of the manuscript.

==============================
Spermatogenesis and sperm maturation are complex and highly ordered biological processes. Any failure or disorder in these processes can cause defects in sperm morphology, motility, and fertilization ability. Cathepsin B (CTSB) is involved in the regulation of a variety of pathological processes. In the present study, we found that CTSB was abundantly expressed in the male reproductive system, however, the specific role of CTSB in regulating spermatogenesis and sperm maturation remained elusive. Hence, we generated Ctsb-/- mice using CRISPR/Cas9 technology. In Ctsb-/- mice, sperm count was significantly decreased while the level of morphologically abnormal sperm was markedly increased. Additionally, these mice had significantly lower levels of progressive motility sperm and elevated levels of immobilized sperm. Histological analysis showed slight vacuolization in the testis epithelium, as well as the loss of epididymal epithelium cells. Further investigation showed that autophagic activity was inhibited and apoptotic activity was increased in both the testis and epididymis of Ctsb-/- mice. Together, our findings demonstrate that CTSB plays an important role in spermatogenesis and sperm maturation in mice.

Introduction

It has been estimated that 72.4 million people worldwide are affected by couple infertility, where the male factor accounts for 50% of couple sub-fertility (Sironen et al., 2020). Male infertility and decreased sperm quality have become a serious global problem (Agarwal et al., 2015). Idiopathic male infertility with no identified etiological factors accounts for approximately 30–40% of male infertility cases (Jungwirth et al., 2012). Defects in the seminiferous tubules affecting spermatogenesis and defects in the epididymis affecting sperm maturation are important causes of male infertility.

CTSB is a member of the cysteine protease family, whose biological function has been extensively studied. Previous research has demonstrated that CTSB plays a key role in many biological functions and human diseases including the turnover of cellular proteins (Chitranshi et al., 2021), angiogenesis (Yanamandra et al., 2004), tumor proliferation (Mijanovic et al., 2019), neurological disease (Cermak et al., 2016; Moon et al., 2016), cholesterol absorption (Zhang et al., 2014) and so on. CTSB activity and function are closely related to autophagic activity (Araujo et al., 2018; Wang et al., 2020b). Overexpression of CTSB has been shown to promote autophagy in cultured IMR-90 cells (Xiao et al., 2018). By restoration of autophagy flux though activation of CTSB and inhibiting ROS/p38/JNK pathway, cilostazol could alleviate nicotine induced cardiomyocytes hypertrophy (Wang et al., 2020b). Also, CTSB regulates the activity of the downstream TOR signalling pathway and initiates the autophagy of breast cancer cells (Han et al., 2017). There is coordination between the processes of autophagy and apoptosis. CTSB mediated degradation of Disabled-2 (Dab2) allows for the induction of autophagy (Jiang et al., 2016) and CTSB inhibition maintains Dab2 expression, while sustained Dab2 expression prevents autophagy and promotes apoptosis by stabilizing the pro-apoptotic Bim protein (Jiang et al., 2016). Other research has found that alantolactone, a pharmacological inhibitor of autophagy in pancreatic cancer cells, inhibits the expression and activity of the CTSB protein and results in the occurrence of apoptosis (He et al., 2018). Also, CTSB plays a critical role in folliculogenesis in female mice by mediating autophagy, apoptosis, and proliferation (Chen et al., 2021).

Evidence indicates that autophagy is involved in many cellular events within the male reproductive system (Zhu et al., 2019). Autophagy is a conserved process of the degradation of impaired or dysfunctional organelles and proteins, which has a key role in the maintenance of spermatogenic intracellular homeostasis and helps improve sperm motility (Zhu et al., 2019). Autophagy happens during spermatogonia and is elevated under an adverse environment (Liu et al., 2015; Xu et al., 2016). The injection of autophagy inhibitors into the testis or knockout of Atg7 in germ cells results in abnormal acrosome biogenesis in mice (Wang et al., 2014). The deletion of autophagy protein ATG5 in male germ cells decreases testicular autophagic activity, causing a significant reduction in both sperm count and motility (Wang et al., 2022). Besides, autophagy regulates cytoskeleton organization, thus facilitating the differentiation of spermatids (Shang et al., 2016). Additionally, proteins related to autophagy have been found in human spermatozoa, where autophagy activation induced a significant increase in motility and autophagy inhibition resulted in decreased motility and viability (Aparicio et al., 2016). Down-regulated expression of autophagy associated genes lead to low sperm quality (Guo et al., 2021), deficient autophagy pathway is found in spermatozoa of individuals with globozoospermia (Foroozan-Boroojeni et al., 2021). Also, the increased autophagic activity is supposed to mitigates the damage to testis and germ cells in infertile men with varicocele (Foroozan-Broojeni et al., 2019) and in a rat model of varicocele (Sadeghi et al., 2020). As these studies confirm that sperm quality is related to autophagy, they further highlight the importance of studying autophagy in the male reproductive system.

CTSB is one of the key molecules of human spermatogenesis and sperm maturation screened in our previous work. In the present study, we found that CTSB is expressed in the male reproductive system. We generated Ctsb knockout mice to study the role of CTSB in autophagy and the male reproductive system.

Material and Methods

Animals

All animal experimental procedures were approved by the Ethics Committee of Shandong First Medical University (W202111230331, Jinan, China). Animal experiments were strictly carried out in accordance with the national and local animal protection laws. All mice were maintained under SPF conditions (22–24 °C, 50–55% humidity,12 h light/dark cycle) with free access to water and food in Laboratory Animal Center of Shandong First Medical University. All mice were treated humanely and with efforts to minimize suffering. To induce loss of consciousness and death with a minimum of pain and distress, all mice were euthanized by cervical dislocation to collected tissue samples for further analyses. There were no surviving animals at the end of study.

Quantitative RT-PCR (RT-qPCR)

Total RNA was extracted using TRIzol reagent (Invitrogen, Carlsbad, CA, USA). First strand cDNA was synthesized from total RNA using Primescript Reverse transcriptase (Takara, Tokyo, Japan). Then, a 10 µl mixture was made up containing 5 µl SYBR Premix Ex Taq reagent system (TakaRa, Tokyo, Japan), 0.2 µl Ctsb forward primer: 5′-ACCTTTGATGCACGGGAACA-3′, 0.2 µl Ctsb reverse primer: 5′-ACTCGGCCATTGGTGTGAAT-3′, 1 µl cDNA template, and 3.6 µl ddH2O. RT-qPCR was performed on a BioRad Sequence Detection System (Bio-Rad Laboratories, Hercules, CA, USA) with usual amplification condition (95 °C for 10min, (95 °C 15 s, 65 °C 30 s, 72 °C 30 s) × 40). Triplicates were performed for each reaction. The mRNA levels were calculated using the 2−ΔΔCT method with glyceraldehyde phosphate dehydrogenase (GAPDH) as internal control, and the Gapdh primers were: forward, 5′-GATGCCCCCATGTTTGTGAT-3′, reverse, 5′-GGCATGGACTGTGGTCATGAG-3′.

Generation of Ctsb-/- mice

Ctsb-/- mice were generated using CRISPR/Cas9 technology. The single-guide RNA (sgRNA) target sequences were 5′-CCCTTGAGCGACAGGAAAAACCA-3′ and 5′-TTTCCAAAATTTAGCGGCCCTGG-3′. The sgRNAs were produced through in vitro transcription using a MEGAshortscript T7 kit (Ambion, Austin, TX, USA). The hCas9 mRNA was derived from pST1374-N-NLS-flag-linker-cas9, which was synthesized using the mMESSAGE mMACHINE T7 kit (Ambion, Austin, TX, USA), and polyadenylated with a Poly(A) Polymerase Tailing kit (Life Technologies, Carlsbad, CA, USA). Wild-type C57BL/6 superovulated females at 5-6weeks of age were mated with adult C57BL/6 males to obtain zygotes. The Cas9 mRNA and sgRNA were injected into zygotes by microinjection. The injected zygotes were then transferred into the oviducts of pseudopregnant CD1 female mice. Genomic DNA was extracted from the tails of newborn founders. The genomic DNA fragments surrounding the sgRNA target sites were amplified by PCR (primers: Ctsb forward, 5′-ATGTAGCACATTCACTCTGTAAGC-3′; Ctsb reverse 1, 5′-CTTTTGGAAGTCCTGCAGTCAAA-3′; and Ctsb reverse 2, 5′-AAAGGGCCATGTTAAATTCCTTCTG- 3′).

Western blotting

Protein was extracted using Pierce™ RIPA Buffer (Thermo Fisher, NY, USA) with a protease inhibitor cocktail (Bimake, Houston, TX, USA). The denatured proteins were subjected to 10% sodium dodecyl polyacrylamide gel electrophoresis (SDS/PAGE) followed by electrotransfer onto a PVDF membrane. After being blocked with 5% skim milk in TBST for 2 h at room temperature, the membranes were incubated overnight with primary antibodies at 4 °C. The antibodies used were as follows: anti-CTSB (1:1,000; Abcam, Cambridge, UK), anti-LC3 (1:1,000; Proteintech, Chicago, USA), anti-ATG5 (1:1,000; Abcam, Cambridge, UK), anti-Caspase3 (1:1,000; Abcam, Cambridge, UK), anti-Cleaved Caspase3 (1:1,000; CST, Boston, USA), anti-β-actin (1:5,000; Abcam, Cambridge, UK), and anti-GAPDH (1:5,000; GeneTex, San Antonio, USA). After being washed with TBST, the membranes were incubated with secondary antibodies for 1 h at room temperature. The bands on Western blotting were quantified using Image J software and normalized to β-actin or GAPDH.

Histological analysis and fertility test

The testes and epididymides were isolated from the adult male mice and fixed with Bouin’s solution for 12 h at room temperature. Next, the tissues were dehydrated via an ethanol series from 30% to 100%. Afterwards, the tissue samples underwent xylene dehydration, clearing, and embedded into paraffin. Tissue sections (4 µm) were spread onto slides and dried overnight at 55 °C. After deparaffination and rehydration by immersing in serial concentrations of ethanol, tissue sections were stained with hematoxylin and eosin (H&E) for histological analysis. Sperm samples from the cauda epididymis were spread onto slides and air dried overnight. Finally, after being fixed with 4% paraformaldehyde in PBS for 30 min, sperm samples were stained with H&E according to standard protocols. At least 200 sperm cells were used to assess the sperm morphology in each mouse, 4–6 mice were evaluated in each group.

To test the reproductive ability, 2-month-old WT or Ctsb-/- males were caged with adult WT females in a ratio of 1:2 for 3 months. The number of pups per litter was recorded. Four mating cages were set for each group.

Computer-assisted sperm analysis (CASA)

The cauda epididymis was detached in each adult male mouse, washed with PBS, and placed in an M2 medium (Sigma-Aldrich, St. Louis, MO, USA). Sperm cells were allowed to exude from incisions of the cauda epididymis for 30 min at 37 °C under 5% CO2, after which any debris was removed and volume was increased to one mL with M2 medium. 10 µL of the exudate was used to assess sperm motility and sperm count by using the CASA system (Tsinghua Tongfang, Beijing, China).

Immunohistochemistry

The tissues were fixed with 4% PFA for 12 h at room temperature, then dehydrated and embedded into paraffin. Immunohistochemistry was performed on paraffin-embedded sections. After deparaffination and rehydration, tissue sections were subjected to antigen retrieval. Endogenous peroxidase was inhibited through incubation with 3% H2O2 for 10 min at room temperature, and tissue sections were blocked in 10% normal goat serum for 30 min at room temperature. Then, the tissue sections were incubated in primary antibodies at 4 °C overnight. The primary antibodies used in this study included a rabbit monoclonal anti-CTSB antibody (1:200, Abcam, Cambridge, UK) and a rabbit monoclonal anti-SOX9 antibody (1:200, ABclonal, Wuhan, China). The staining process was performed using a Streptavidin-Peroxidase-Biotin kit (SP-9000, ZSGB-BIO, Beijing, China), and chromogenic reaction was conducted with a DAB kit (ZLI-9017; ZSGB-BIO, Beijing, China) according to the manufacturer’s instructions.

Immunofluorescence staining

Immunofluorescence staining was performed on paraffin-embedded sections as previously described (Wen et al., 2020). Specifically, after deparaffination and rehydration, tissue sections were subjected to antigen retrieval and permeabilization. Then, the sections were blocked with 5% normal goat serum and incubated with primary antibody (γ-H2AX; Abcam, Cambridge, UK) at 4 °C overnight and secondary antibody at 37 °C for 1 h. The nuclear was stained with DAPI, and the images were captured using ZEISS LSM 880 confocal laser scanning microscope (ZEISS, Oberkochen, Germany).

The FITC conjugate Arachis hypogaea (peanut) agglutinin (PNA) was used to assess the sperm acrosome. The sperms collected from cauda epididymis were carefully washed with PBS, then coated on slides and air-dried. The samples were fixed with 4% PFA, washed with PBS, and stained with PNA at 37 °C for 30 min. For the acrosome staining in testis, testicular paraffin sections were dewaxed and rehydrated, then PNA was stained at 37 °C for 30 min.

Transmission electron microscopy (TEM)

The caput epididymides were rapidly isolated from WT and Ctsb-/- mice and fixed with 2.5% glutaraldehyde overnight at 4 °C. After post-fixation with 1% osmic acid, the caput epididymides were infused in 10% gelatin, dehydrated in sucrose, and then frozen in liquid nitrogen. Cryosections (50 nm) were prepared using a cryo-ultramicrotome (Leica, Wetzlar, Germany) and observed with a JEM-1200EX microscope (JEOL, Tokyo, Japan) following the manufacturer’s protocol.

TUNEL assay

Cell apoptosis was measured using terminal deoxynucleotidyl transferase-mediated dUTP-biotin nick-end labeling (TUNEL) assay with an in situ cell death detection kit (Roche, Basel, Switzerland) following the manufacturer’s protocols.

Statistical analysis

All statistical data were analyzed by using GraphPad Prism, RT-qPCR data and Western blot data were normalized to GAPDH or β-actin. Data were expressed as mean ± standard deviation (SD). The Student’s t test was used for data comparison, and P < 0.05 was considered statistically significant.

Results

Expression analysis of Ctsb in mice

We adopted an RT-qPCR assay to elucidate the relative expression levels of Ctsb in different mouse tissues. As the results show, Ctsb demonstrated ubiquitous expression in mouse tissues (Fig. 1A). In the male reproductive system, a small amount of Ctsb was expressed in the testis, but Ctsb was abundantly expressed in the caput epididymis and cauda epididymis. Next, we used immunohistochemistry to further confirm the expression levels of CTSB in the male reproductive system. Consistent with the results of RT-qPCR, CTSB was highly expressed in both the caput epididymis and cauda epididymis, and a small amount of CTSB was present in the testes mainly in Leydig cells (Fig. 1C). Furthermore, we found that CTSB was primarily expressed in the tail of spermatozoa (Fig. 1B). The abundant expression of CTSB in spermatozoa and the male reproductive system implied that CTSB might participate in spermatogenesis or sperm maturation.

Figure 1 Expression of Ctsb in mouse tissues.

(A) Expression analysis of Ctsb in mouse tissues by RT-qPCR. Tissues included the heart, liver, spleen, lung, kidney, caput epididymis (caput), cauda epididymis (cauda), testis, brain, stomach, colon, small bowel, womb, and ovary. Triplicates were performed for each reaction. (B) Immunofluorescence detection of CTSB location in mouse spermatozoa. (C) Immunohistochemistry signals show the locations of CTSB in the testis, caput epididymis, and cauda epididymis of WT mice; arrows indicate the Leydig cells. CTSB was undetectable in the testis and epididymis of Ctsb-/- mice.

Generation of Ctsb-/- mice using CRISPR/Cas9

To study the function of CTSB in male reproduction, CRISPR/Cas9 was used to knockout the Ctsb gene. We designed two sgRNAs that were used to delete a large portion of the coding sequence of Ctsb (Fig. 2A). The resultant deletion was confirmed by sequencing and PCR (Figs. 2B and 2C). Western blotting showed no band at the expected size for the CTSB protein in Ctsb-/- mice (Fig. 2D), and CTSB was undetectable by immunohistochemistry analyses in the homozygote testis and epididymides of Ctsb-/- mice (Fig. 1C). These results consistently indicated that Ctsb-/- mice were successfully generated. To test the fertility of the Ctsb-/- male mice, adult WT and Ctsb-/- male mice were mated with adult WT female mice at a ratio of 1:2 for three months. The number of pups from each litter was recorded, and these results showed a smaller average number of pups for Ctsb-/- male mice (Fig. 2E).

Figure 2 CRISPR/Cas9 mediated generation of Ctsb-/- mice.

(A) Schematic representation of targeting strategy using CRISPR/Cas9 system. Two sgRNA target sites were indicated, and the deletion between the sgRNA target sites was identified by the forward (F) and reverse (R1, R2) primers. (B) DNA sequencing chromatograms showing a 9819bp deletion in Ctsb-/- mice. (C) Agarose gel electrophoresis analysis showing DNA bands for the different genotypes. (D) Western blot showing no band at the expected size of the CTSB protein in Ctsb-/- mice. (E) Fertility test showing the number of pups per litter from four WT and four Ctsb-/- males. Each black dot indicates the number of mice born in each litter. * P < 0.05 vs WT.

Impaired sperm quality in Ctsb-/- mice

To investigate the function of CTSB in the male reproductive system, we paid close attention to the development of the testis and epididymis in Ctsb-/- mice. No significant differences in weight and morphology of the testis, epididymis, or seminal vesicle were found between adult WT and Ctsb-/- mice (Figs. 3A and 3B). Histology analysis showed slight degeneration of the seminiferous tubules characterized by vacuolization of the epithelium in Ctsb-/- mice (Fig. 3C). Additionally, Ctsb-/- mice had a decreased number and sparse arrangement of epithelial cells in the caput epididymis (Fig. 3C). Next, we analyzed whether the processes of meiosis and spermiogenesis were normal in Ctsb-/- mice. The expression of γH2AX was used to identify the meiotic process, whereas PNA was used to identify acrosome development. Immunofluorescence analysis showed a normal presence of γH2AX-positive spermatocytes (Fig. 3D), PNA staining showed normal development of acrosome in the round spermatids at stage VII and VIII (Fig. 3E) of Ctsb-/- mice. Ctsb-/- mice also had normal levels of SOX9-positive Sertoli cells (Figs. 3F and 3G).

Figure 3 Spermatogenesis of Ctsb-/- mice.

(A) The morphology of the testis, epididymis, and seminal vesicle in WT and Ctsb-/- mice. (B) The weights of the testis, epididymis, and seminal vesicle did not significantly differ between WT and Ctsb-/- mice. (C) H & E-stained sections of the testes, caput, and cauda epididymides from WT and Ctsb-/- mice. (D) Immunofluorescence staining of γH2AX-positive spermatocytes from WT and Ctsb-/- mice. (E) PNA-stained acrosomes in the developmental spermatid of WT and Ctsb-/- mice. (F) Immunohistochemistry staining of SOX9-positive Sertoli cells from WT and Ctsb-/- mice. (G) Sertoli cell counts from WT and Ctsb-/- mice. NS, non-significant. n ≥ 4 in each group.

Next, sperm quality, including morphology and motility, was examined in Ctsb-/- mice. H&E staining showed that morphologically abnormal sperm was increased in the cauda epididymis of Ctsb-/- mice (Fig. 4A). The morphologically abnormal sperm were compared in testis, caput epididymis, and cauda epididymis between WT and Ctsb-/- mice (Fig. 4B). The ratio of morphologically abnormal sperm in testis and caput epididymis was significantly increased in Ctsb-/- mice. However, the ratio of morphologically abnormal sperm in cauda epididymis was more pronounced in Ctsb-/- mice. Specifically, the proportions of sperm cells with abnormal heads, coiled tails, and that were decapitated were all markedly increased in the cauda epididymis of Ctsb-/- mice (Fig. 4C). Meanwhile, PNA-staining showed that Ctsb-/- mice had a higher percent of sperm cells with abnormal acrosome morphology (Figs. 4D and 4E). Compared to WT mice, the sperm count was decreased in Ctsb-/- mice (Fig. 4F). CASA analysis was used to evaluate sperm motility. The proportion of progressive motility sperm was decreased, and the proportion of immobilized sperm was elevated in Ctsb-/- mice (Fig. 4G). The increased level of morphologically abnormal sperm and attenuated sperm motility suggested that sperm quality was impaired in Ctsb-/- mice.

Figure 4 Histology and sperm quality analysis of Ctsb-/- mice.

(A) H & E-stained sperm cells from WT and Ctsb-/- mice, where morphologically abnormal sperm is indicated by the arrows. (B) The ratio of morphologically abnormal sperm in the testis, caput epididymis (caput) and cauda epididymis (cauda) of WT and Ctsb-/- mice. (C) The ratios of sperm cells with abnormal heads, coiled tails, and that were decapitated in the cauda epididymides of WT and Ctsb-/- mice. (D) PNA-stained acrosomes in the sperm cells of WT and Ctsb-/- mice. (E) The ratio of morphologically abnormal acrosomes in the sperm cells of WT and Ctsb-/- mice. (F) Sperm counts in the cauda epididymides of WT and Ctsb-/- mice. (G) CASA analysis of sperm cells from the cauda epididymides of WT and Ctsb-/- mice. PR, progressive motility; NP, non-progressive motility; IM, immobilized sperm. * P < 0.05 vs WT , ** P < 0.01 vs WT, NS, non-significant. n ≥ 4 in each group.

Dysregulated autophagic and apoptotic activity in Ctsb-/- mice

To study the mechanism of sperm quality damage in Ctsb-/- mice, we investigated autophagic activity in the male reproductive system. The expression levels of ATG5, an autophagy protein with a core role in germ cell development, and autophagy-associated gene LC3, a widely used marker for mammalian autophagy, were analyzed by Western blotting. Our results suggested that autophagy was suppressed in the testis of Ctsb-/- mice, as indicated by the decreased protein levels of ATG5 and LC3II/LC3I (Figs. 5A and 5B). As consistent with the observations in testis, ATG5 expression and the ratio of LC3II/LC3I were markedly decreased in the epididymis of Ctsb-/- mice (Figs. 5C and 5D). Immunohistochemistry staining of LC3 and ATG5 also showed decreased autophagic activity in the testis and caput epididymis of Ctsb-/- mice (Fig. 5E). Next, we used TEM to examine the ultrastructural autophagic processes in the epididymis of WT and Ctsb-/- mice. We found that the number of autophagic vacuoles surrounded by double-membrane structures was significantly decreased in Ctsb-/- mice (Fig. 5F). Ctsb -/- mice also had a significant number of epididymis cells with apoptosis changes, which mainly involved chromatin condensation and distributes along the nuclear membrane (Fig. 5F). The TUNEL assay was used to assess the apoptotic activity in the male reproductive system of WT and Ctsb-/- mice. Our data revealed that, compared to WT mice, the number of TUNEL-positive cells was significantly increased in both testis and epididymis of Ctsb-/- mice (Fig. 5G). Also, Western blotting results showed that the level of cleaved-caspase3 was elevated in both testis (Fig. 5H) and epididymis (Fig. 5I) of Ctsb-/- mice, further indicating increased apoptotic activity. Together, these observations would suggest that Ctsb knockout inhibits autophagic activity and promotes apoptotic activity in the male reproductive system of mice.

Figure 5 Autophagic and apoptotic activity detected in Ctsb-/- male mice.

(A, B) Western blotting used to detect the protein expression levels of ATG5 and LC3II/LC3I in the testis of WT and Ctsb-/- mice. (C, D) Protein expression levels of ATG5 and LC3II/LC3I in the epididymis of WT and Ctsb-/- mice. (E) Immunohistochemistry staining of LC3 and ATG5 in the testis and caput epididymis from WT and Ctsb-/- mice. Scale bar = 50 µm. (F) Autophagic and apoptotic processes of epididymis cells examined by TEM in WT and Ctsb-/- mice, autophagosomes were indicated. Scale bar = 5 µm. (G) TUNEL assay was used to detect the apoptotic activity in the testis and epididymis of WT and Ctsb-/- mice. Scale bar = 100 µm. (H) Western blot used to detect cleaved-Caspase3 in the testis of WT and Ctsb-/- mice. (I) Western blot used to detect cleaved-Caspase3 in the epididymis of WT and Ctsb-/- mice. * P < 0.05 vs WT, ** P < 0.01 vs WT, NS, non-significant, n ≥ 4 in each group.

Discussion

The ongoing efforts to understand the mechanisms essential for healthy spermatogenesis and sperm maturation are of great significance, as they could help explain the basis of male factor infertility and provide possible diagnosis and treatment approaches for advancing male reproductive health. In this study, we generated a Ctsb knockout mouse model to investigate the function of CTSB in the male reproductive system for the first time. In the Ctsb-/- mice, we observed a decreased sperm count, lower sperm motility, and increased presence of morphologically abnormal sperm in the cauda epididymis. Defects in spermatogenesis and sperm maturation could be concurrent in the Ctsb-/- mice.

Spermatogenesis occurs in the seminiferous tubules, and an ample array of factors can influence the quality of this process (Wu et al., 2020). The sperm count was decreased and the ratio of morphologically abnormal sperm in testis was increased in Ctsb-/- mice. Additionally, in the testis of Ctsb-/- mice, slight vacuolization was found in the seminiferous tubules. Further investigation revealed that autophagic activity was decreased and apoptotic activity was increased in the testis of Ctsb-/- mice, which could explain at least part of the vacuolization in the testes. These defects were highly correlated with abnormal spermatogenesis in the testes. In the process of sperm maturation, sperm cells passes through the epididymis to acquire both fertilizing ability and forward motility properties  (Sullivan & Mieusset, 2016; Dun, Aitken & Nixon, 2012). Issues with sperm maturation in the epididymis can cause an increase in abnormal sperm morphology and defects in sperm motility  (Nanjappa et al., 2016; Joseph et al., 2010; Luo & Sun, 2013). In the present study, CTSB was found to be highly expressed in the epididymis. Autophagy was repressed in the epididymis of Ctsb-/- mice. Additionally, Ctsb knockout promoted apoptosis and caused a decreased number and sparse arrangement of epithelial cells in the epididymis. The loss of epithelial cells could then cause the deterioration of the epididymal microenvironment, which is detrimental to sperm maturation. In the cauda epididymis of Ctsb-/- mice, the increased ratio of morphologically abnormal sperm compared with caput epididymis indicated defective sperm maturation in the epididymis. Attenuated sperm motility is also related to the defect of sperm maturation in the epididymis. As a cysteine protease, CTSB may also exist in germ cells or sperm acrosomes and play an important role in the structure or biogenesis of sperm acrosomes. PNA staining also showed elevated abnormal acrosome in the Ctsb-/- sperm cells. Ultimately, defects in spermatogenesis and sperm maturation led to impaired sperm quality.

Autophagy and apoptosis are essential for the homeostasis of organisms, where the balance between autophagy and apoptosis is vital for the fate of cells. Apoptosis, the confirmed genetic programmed death process, has been extensively studied and its contribution to disease pathogenesis is well documented. Studies have shown that autophagy has a complex interaction with apoptosis (Eisenberg-Lerner et al., 2009; Zhuang et al., 2018). Autophagy usually blocks the induction of apoptosis, whereas the activation of apoptosis-related caspases turns off the autophagy process. Cross-talk between autophagy and apoptosis regulates testicular injury and recovery (Wang et al., 2020a). Rapamycin inhibits spermatogenesis and reduces sperm count through changing the status of autophagy and apoptosis (Liu et al., 2017). Sitagliptin attenuates the cadmium-induced testicular impairment by activating autophagy and inhibiting apoptosis (Arab et al., 2021). In the current study, Ctsb knockout inhibited autophagy and promoted apoptosis in the testes and epididymis. This relationship between autophagy and apoptosis was consistent with the previously mentioned mechanism, i.e., if a genetic defect or pharmacological inhibition blocks one program, the other takes over (Eisenberg-Lerner et al., 2009). We observed impaired sperm quality in Ctsb-/- mice; however, the studies on the mechanism of CTSB deficiency leading to impaired sperm quality are insufficient. A better understanding of the mechanisms that regulate autophagy and apoptosis is essential for discovering of therapeutic tools in the strenuous fight against male infertility. Our findings further elucidate the role of CTSB in regulating autophagy and apoptosis and can be used to inspire prospective strategies for male infertility therapy.

Conclusions

Fertility is reduced in Ctsb-/- male mice. Ctsb deficiency causes spermatogenesis and epididymal sperm maturation defects. Autophagic and apoptotic activity are disordered in Ctsb-/- male mice.

Supplemental Information

Supplemental Information 1 Expression analysis of Ctsb in mouse tissues by RT-qPCR

Click here for additional data file.

Supplemental Information 2 Fertility test showing average pups per litter from WT ( n = 4) and Ctsb-/- ( n = 4) male mice

Click here for additional data file.

Supplemental Information 3 Western blot showing no band at the expected size of the CTSB protein in Ctsb-/- mice

Click here for additional data file.

Supplemental Information 4 The weights of the testis, epididymis, and seminal vesicle did not significantly differ between WT and Ctsb-/- male mice

Click here for additional data file.

Supplemental Information 5 Sertoli cell counts from WT and Ctsb-/- male mice

Click here for additional data file.

Supplemental Information 6 The ratio of morphologically abnormal sperm in the cauda epididymides of WT and Ctsb-/- male mice

Click here for additional data file.

Supplemental Information 7 The ratios of sperm cells with abnormal heads, coiled tails, and that were decapitated in the cauda epididymides of WT and Ctsb-/- male mice

Click here for additional data file.

Supplemental Information 8 The ratio of morphologically abnormal acrosomes in the sperm cells of WT and Ctsb-/- male mice

Click here for additional data file.

Supplemental Information 9 Sperm counts in the cauda epididymides of WT and Ctsb-/- male mice

Click here for additional data file.

Supplemental Information 10 CASA analysis of sperm cells from the cauda epididymides of WT and Ctsb-/- male mice

Click here for additional data file.

Supplemental Information 11 Raw data for western blot in Figures 2 and 5

Click here for additional data file.

Supplemental Information 12 checklist

Click here for additional data file.

Additional Information and Declarations

Competing Interests

Author Contributions

Data Availability

The authors declare there are no competing interests.

Zongzhuang Wen performed the experiments, prepared figures and/or tables, and approved the final draft.

Haixia Zhu performed the experiments, prepared figures and/or tables, and approved the final draft.

Bin Wu analyzed the data, prepared figures and/or tables, and approved the final draft.

Aizhen Zhang analyzed the data, prepared figures and/or tables, and approved the final draft.

Hongxiang Wang performed the experiments, prepared figures and/or tables, and approved the final draft.

Yin Cheng performed the experiments, prepared figures and/or tables, and approved the final draft.

Hui Zhao performed the experiments, prepared figures and/or tables, and approved the final draft.

Jianyuan Li conceived and designed the experiments, authored or reviewed drafts of the article, and approved the final draft.

Min Liu analyzed the data, authored or reviewed drafts of the article, and approved the final draft.

Jiangang Gao conceived and designed the experiments, authored or reviewed drafts of the article, and approved the final draft.

The following information was supplied regarding data availability:

The raw data are available as Supplemental Files.

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
