# Peer review of "Cathepsin B plays a role in spermatogenesis and sperm maturation through regulating autophagy and apoptosis in mice"

_PeerJ, doi:10.7717/peerj.14472_

## Round 0.1 · original submission · Major Revisions

External reviewers agree on the novelty of the study and that it warrants publishing. However, there are some outstanding issues that need addressing or clarification. Please consider all comments carefully and respond accordingly in a revised version. Also, it would be really valuable, as pointed out by Rev 1 to report body weights. Perhaps consider reporting both total and adjusted (weight/bw) testis weights.

Reviewer 1 ·

Basic reporting

This study determined that ctsb deletion affects testicular and epididymal development and assessed the role of ctsb gene deletion leading to male reproduction by constructing knockout mice using crispr-Cas9 technology. This is interesting work, but there are still some flaws and some suggestions:
1. Compared with testis and epididymis, the mRNA expression of Ctsb was higher in the liver, spleen and uterus in figure 1A. Why did the follow-up study focus on the testis and epididymis?
2. Shoeb Ikhlas et al (Chemosphere, 2019) and Grigor Varuzhanyan et al(elife, 2019)have shown that the analysis you use in Lines 139-145 is not the most appropriate for this situation. Please explain why you used this method. In addition, relevant results such as sperm counts (at least 60x106) differ from those already reported (no more than 12x106).
3. Although the results of the current study showed no significant difference in testicular and epididymal weights between the control and treated groups, testicular and epididymal weights are closely related to body weight. Please provide data on the body weight of the mice.
4. What is the scale bar in figure5 E?
5. What is the litter size of their offspring? in line 439, “Fertility test showing average pups per litter from WT (n = 4) and Ctsb-/- (n = 4) male “, what does n mean? Also, in Figure 2E, what does the black dot indicate?
6. With CTSB knockout by crispr-cas9 technology, it is difficult to determine whether sperm abnormalities occur in the testicular and thus in the epididymis, or whether they are a direct result of Ctsb deletion in the testis and epididymis. If testis- or epididymis-specific knockout Ctsb mice were used, it would be possible to accurately determine whether the testis or epididymis deletion of Ctsb was responsible for sperm abnormalities. At least, the sperm abnormalitites should be compared between the sperm from caput epididymis and cauda epididymis.
7. The results of apoptosis and autophagy are of great interest, but the evidence is slightly insufficient. The electron microscopy results show a decrease in autophagy after the knockdown of ctsb. If possible, autophagy staining was performed to verify the electron microscopy results; the fluorescence signal of apoptosis staining was also increased, suggesting that the apoptosis rate should be counted.
8. The authors should carefully revise the language in the manuscript.

Experimental design

no comment

Validity of the findings

no comment

Additional comments

no comment

Reviewer 2 ·

Basic reporting

A manuscript by Wen et al. has described the role of a cysteine protease, cathepsin B (CTSB), in spermatogenesis and sperm maturation in mice. In general, the author showed expression of CTSB in male reproductive system, both testis and epididymis. They successfully generated CTSB knock out mice (Ctsb-/-) using CRISPR/Cas9 technology, which was proven by PCR, Western blot analysis. The knocked-out CTSB did not affect overall features of male reproductive organs but the morphology (head, tail and the acrosome) and motility of epididymal sperm were significantly altered compared to those taken from wild type mice. The autophagic activity of both testis and epididymis as measured through the gene markers are reduced while many apoptotic signs were enhanced. Together, the authors claimed that CTSB plays an important role in spermatogenesis and sperm maturation. Although the manuscript is well written and well organized to ease understanding, however, there are some points remain to be improved prior to its acceptance for publication.
Title:
(Line 1) – The results present in this MS did not directly support involvement of CTSB in any step of spermatogenesis or sperm maturation in epididymis. The authors showed more about the effect of CTSB knockout towards the disturbance of autophagy and apoptosis interplay, which then affects sperm morphology and thus motility. The title thus needs to be rephrased to reflect the highlighted results that they obtained.
Introduction:
(Lines 53-60) – I didn’t find this part of Introduction useful, but rather lengthy. Also in the second paragraph (Lines 61-69), the involvement of CTSB in cancer progression and other biological functions can be shortened as the information is not so related. I would look for more information about the involvement of CTSB or any other enzymes in autophagy/apoptosis interplay (which are quite extensively mentioned in Discussion). Please rephrase these 2 paragraphs of Introduction to make it more concise which will guide the readers to understand your work better.
(Lines 95-96): The reference is needed for the presence of CTSB in human reproduction. Indeed, this will be a good clue for the authors to extrapolate the function of CTSB by generating CTSB-KO mice.

Experimental design

Materials and Methods:
(Lines 111-120) – The writing in this method is confusing and lacks some information. First, the sequences of primers for GAPDH should go to Line 120 to explain GAPDH as internal control. Second, the PCR amplification condition should be given.
(Line 112 and others) – the reference of company for each chemical should be consistent. The format in Line 128 is more standard (with the name of city).
(Line 130) – Cas9 mRNA and sgRNA injection? I assume that the authors injected these RNA molecules (or vectors?) into zygotes by microinjector. If that is the case, please extrapolate your writing to make the method clear. Also, the testing results of KO-CTSB are relied both on PCR and Western blot (as shown in Fig. 2). However, the part of Western blotting is absent here, should it be referred to session 2.9?
(Lines 135-156) Methods of sessions 2.4 and 2.6 should be combined.

Validity of the findings

Results:
(Lines 222-250) – As mentioned above that overall features of testis and epididymis of CTSB-/- were not affected and still comparable to wild type (Fig. 3), whereas the sperm morphology and motility showed many significant changes. The major concern is should these sperm abnormalities are taken place as early as in testis or in epididymis? If possible, I would expect to see similar results of Figure 4 in the testicular sperm in addition to only epididymal sperm, so that a firm conclusion can be made about the site where sperm morphology is defective.
(Lines 230-231 and Line 259) – The authors have mentioned so many methods to show the expression of H2AX, the staining of PNA, and the TEM analysis, I didn’t see any relevant methods of these results. Please add these methods into Materials and Methods where applicable.

Discussion:
Expression of CTSB in this study appeared to be more prominent in Leydig cells (testis) or epithelial origin (epididymis). Although some discussions have already made about the non-germ cell CTSB that may be involved in autophagy-apoptosis pathways in either testis or epididymis which then cause the altered microenvironment in male reproductive system and sperm cell structures thereafter. As a rather universal house-keeping enzyme, germ cell inherent CTSB may also exist, particularly in the acrosome of sperm (if any). The authors should mention the possibility of sperm inherent CTSB and its feasibility to be involved in abnormal acrosome structure or biogenesis in the sperm cells.

Additional comments

N/A

---

## Round 0.2 · Minor Revisions

The authors have addressed all issues raised by external reviewers in an acceptable manner. The manuscript is ready to be accepted for publication. However, there was one final comment made to Fig panel 5I, with reference to Caspase 3 WB images. Are they the correct ones? Once this final comment has been addressed adequately, the manuscript can progress to publishing.

Reviewer 1 ·

Basic reporting

The authors addressed my concerns.

Experimental design

No comment

Validity of the findings

no comments.

Reviewer 2 ·

Basic reporting

The authors have now responded to my comments and (other reviewer) thoroughly and have comprehensively revised manuscript, so that it is in a good, publishable form. In brief, they are now emphasizing on knocking down of cathepsin-B (ctsb-/-) which affects both autophagy interplayed with apoptosis, thus in turns, affects spermatogenesis in testis and sperm maturation in epididymis. Many figures have been revised to give supporting evidence of the claim which also allows a better understanding for readers.

Experimental design

Materials and Methods have been revised as suggested and have given enough details to follow.

Validity of the findings

As mentioned above, the authors have provided more evidence to support the claim, so that the role of ctsb in autophagy and apoptosis during spermatogenesis and sperm maturation can be claimed. They have added more evidence in testis for ctsb-/- (in Fig. 4 and 5), so that the results give a stronger support for their claim. The MS revision in both introduction and discussion are sound and give a better correlation to the findings.
However, a minor correction needed (which can be revised at the editorial or in-press level) is the raw data of western blot for Fig. 5(I) - caspase3 : the 2 bands shown are either inverted between WT and ctsb-/- or not in order. It doesnot affect much to the ratio in the right panel. Please do check it more carefully.

---

## Round 0.3 · accepted · Accept

With the final minor correction, the authors have addressed all issues raised during peer-review adequately. The manuscript is ready for publication.